# Investigating the Long-Term Variation Trends of Absorbing Aerosols over Asia by Using Multiple Satellites

**Ding Li** [1,2] , **Yong Xue** [1,\*], **Kai Qin** [1], **Han Wang** [1], **Hanshu Kang** [1] **and Lizhang Wang** [1]

1   School of Environment and Spatial Informatics, China University of Mining and Technology, Xuzhou 221116, China
2   Observation and Research Station of Jiangsu Jiawang Resource Exhausted Mining Area Land Restoration and Ecological Succession, China University of Mining and Technology, Ministry of Education, Xuzhou 221116, China
\*   Correspondence: yxue@cumt.edu.cn

**Abstract:** Absorbing aerosols, consisting of smoke (black carbon (BC) and other organics) and dust (from windblown sources), can have a strong warming effect on the climate and impact atmospheric circulation due to localized heating. To investigate the spatiotemporal and vertical changes of absorbing aerosols across Asia, collocation data from OMI, MODIS, and CALIPSO were used to compare two periods: 2006–2013 and 2014–2021. This study revealed a significant temporal and spatial contrast of aerosol loading over the study region, with a drop in total aerosol concentration and anthropogenic smoke concentration recorded across the Eastern China region (all seasons) and a concurrent increase in the Indian sub-continent region (especially in autumn). The range of aerosol diffusion is affected by the height of the smoke and aerosol plumes, as well as the wind force, and is dispersed eastwards because of the Hadley circulation patterns in the Northern Hemisphere. Smoke from Southeast Asia typically rises to a height of 3 km and affects the largest area in contrast to other popular anthropogenic zones, where it is found to be around 1.5–2 km. The dust in Inner Mongolia had the lowest plume height of 2 km (typically in spring) compared to other locations across the study region where it reached 2–5 km in the summer. This study showed, by comparison with AERONET measurements, that combining data from MODIS and OMI generates more accuracy in detecting aerosol AOD from smoke than using the instruments singularly. This study has provided a comprehensive assessment of absorbing aerosol in Asia by utilizing multiplatform remote-sensed data and has summarized long-term changes in the spatiotemporal distribution and vertical structure of absorbing aerosols.

**Keywords:** AAOD; OMI; MODIS; CALIPSO; SMOKE

## 1. Introduction

Absorbing aerosols mainly include black carbon (BC) and organic carbon (OC) from anthropogenic emissions (such as the combustion of fossil fuels and biomass burning, which is generally known as smoke) [1–3] and mineral dust from natural sources (such as the change in land surface due to wind erosion) [4,5]. Compared with the scattering properties of total aerosols [6,7], the common properties of aerosols also affect cloud formation and precipitation [8–10]. However, the biggest difference in absorbing aerosols is the positive radiative forcing, a "warming" effect on the atmosphere similar to greenhouse gases, resulting in the urban heat island effect and uncertainty in the climate system [11,12]. Another important point is that absorbing aerosol (smoke) interacts with human activities, including health, policy, and economic development [13–16]. Therefore, it is important to understand the long-term spatiotemporal change patterns of absorbing aerosols that consider regions undergoing different development pathways.

Recent studies have used observations of single or multiple ground-based stations to effectively obtain the microphysical properties, temporal variation characteristics, and direct

radiative effect of local absorbing aerosols [17–20]. However, the sites for absorbing aerosol are inadequate at both regional and global scales for continuous spatiotemporal changes because of the uneven distribution and monitoring period of sites. Satellite remote sensing is an important way to study the spatiotemporal aerosol distribution with wide-ranging coverage [21]. The research based on MODIS provides long-term stable Aerosol Optical Depth (AOD) products of multiple scales and is widely used in various fields [22–24]. AVHRR's AOD products have data from nearly 40 years, dating back to 1982 [25,26]. Himawari-8 has recently enhanced time resolution to hours or even 10 min [27,28]. The total aerosol concentration's time scale and spatial precision are adequate for various requirements. However, due to the complex microphysical properties of aerosols, the retrieval of absorbing aerosols of specific components is more difficult than the total magnitude (AOD) for passive, visible, multispectral satellites. Multi-angle, polarized satellite sensors, such as MISR [29,30] and POLDER [31,32], use modern methods for Fine Mode Fraction (FMF), Single Scatter Albedo (SSA), and the retrieval of other aerosol properties. However, these satellite algorithms are intricate and provide varying coverages. The continuous backscattering coefficient from active lidar is more effective in detecting absorbing aerosols [33]. The aerosol vertical profile dataset from CALIPSO is widely used in almost all prior hypotheses for satellite aerosol retrieval algorithms and climate model simulations [34,35]. However, due to the linear footprint of this active sensor, it is difficult to make an effective comparison without a coverage scale.

To obtain characteristics such as long-term spatiotemporal and vertical distribution of absorbing aerosols at regional or global scales, some passive satellites equipped with the ultraviolet spectrum, such as OMI or TROPOMI—which are sensitive to the absorption of aerosols that rely on the enhancement of molecular scattering for optical transmission path—have also proven to be useful [36–38]. Furthermore, the vertical height shift connected to aerosol movement is far more significant than the horizontal direction. Adams et al. [39] and Guo et al. [40] developed a scheme to calculate the average Frequency of Occurrence (FoO) at the average vertical height for different aerosol types over several years to reveal the transport process of dust on a large scale. This provides a good idea for studying the vertical pattern change of aerosols. However, in comparison with the ground-based measurements, the AAOD from OMI was somewhat underestimated [41–43]. A better way is to use multiple satellite datasets for simple comparison or a deep combination.

On the regional scale, Asia is the world's largest absorbing-aerosol-emission hotspot, where aerosols have complex seasonal physical characteristics from multiple sources [44,45]. Continuous research includes ground-based observation, to the use of satellites to determine hotspots, as well as using the synergy of multiple satellites and ground-based observation over the local regions and seasons where the high frequency and high magnitude of absorbing aerosols occur. Based on A-Train multi-source satellite data, Wang et al. [46] described the temporal and spatial distribution of dust in northern China using the absorbing aerosols index AAI from 2006 to 2011. Kang et al. [47] investigated the spatiotemporal distribution of absorbing aerosols from 2005 to 2016 using the Ozone Monitoring Instrument (OMI) and found that the optical depth of aerosol absorption (AAOD) shows significant upward trends relating to humid and dry regions. Based on OMI data during 2008–2017. Zhao et al. [48] studied the spatiotemporal distribution of absorbing aerosols in the Yangtze River Delta in the past 10 years and concluded that the interannual variation of AAOD increased first and then decreased in the Yangtze River Delta. In addition, layer height is important when studying aerosols' spatiotemporal changes and transmission paths. Ali et al. [49] classified the aerosol types over Saudi Arabia from 2004 to 2016, including multi-source data such as OMI, CALIPSO, and AERONET, and found that the dust type was dominant both annually and seasonally. Vadrevu et al. [50] explored the relationships between the satellite-retrieved fire counts (FC) and aerosol using multi-satellite datasets at a daily time-step covering ten different biomass-burning regions in Asia. Joshi et al. [51] investigated the BC using ground-based and CALIPSO measurements in the semi-urban

areas of the Ganges plain and showed that the daily variation range and daily variation rate were the highest in winter, followed by autumn, and the lowest in summer.

Many of the above studies have analyzed the spatial and temporal distribution, vertical structure, and transport process of absorbing aerosols, which are crucial to studying aerosol composition as well as regional and global climate. Due to the complexity of absorbing aerosols, the seasonal variation trends in the same latitude and similar areas are different or even opposite, and there is a lack of a comprehensive understanding; most studies, however, only concentrate on a single aerosol type and are restricted to a relatively small region. Meanwhile, OMI has made an important contribution to studying the spatiotemporal distribution of absorbing aerosols due to its sensitive UV band and long-term dataset. However, in some studies, OMI AAOD is generally underestimated when validated using ground-based observations—by introducing AOD, a vertical profile with higher accuracy from other products can effectively improve the accuracy of the final AAOD [52–54]. In addition, aerosol research has been carried out gradually over the past 20 years, but this time span has been marked by serious pollution and is closely related to economic development. In areas with high aerosol incidence, such as China and India, a series of studies have been conducted to analyze and prevent the prevalence of aerosols [55–58]. However, there is no systematic study on how absorbing aerosols have developed in the past 20 years, how the concentration and vertical structure of hotspots have changed, and what factors are related to their spatial and temporal distribution characteristics.

In this study, OMI, MODIS CALIPSO, and ground-based observations from 2006 to 2021 are used to explore the spatiotemporal distribution and vertical structure of absorbing aerosols (including dust and smoke). The long-term study involves two 8-year periods (2006–2013 and 2014–2021, which is also the dividing line for the implementation of the Chinese Clean Air Action Plan [14,15]) for comparison before and after to show the overall appearance after excluding accidental errors. Another highlight is the use of ground-based and CALIPSO data to focus on the spatial-temporal correlation and vertical distribution of hotspots. The first chapter of this paper describes the research background and significance, the second chapter introduces the relevant data and methods, and the third chapter provides the results.

## 2. Data and Methods

### 2.1. Data

In this study, active and passive satellite sensors OMI, MODIS, and CALIPSO products, as well as ground-based AERONET and SONET measurements, were used.

OMI is a hyperspectral sensor installed on Aura satellite with a wavelength range of 270–500 nm, and its spatial resolution at the nadir point is $13 \times 24$ km$^2$ [59]. Its observations are used to retrieve the aerosol properties of ocean and land through OMAERO and OMAERUV algorithms [36,60]. AOD, AAOD, and SSA at 354, 388, and 500 nm are the main results from land OMAERUV near-UV products. Among them, the parameters at 500 nm are derived from the other two bands and are the most commonly used in the studies.

MODIS is a multispectral sensor onboard Terra (launched in December 1999, transits at 10:30 a.m. local time) and Aqua (launched in May 2002, transits at 1:30 p.m.). The processing system of MODIS provides several aerosol products, such as MOD/MYD04 from the Dark Target (DT) and Deep Blue (DB) algorithms [61,62]. The AOD dataset at 440 and 550 nm is one of the most widely used and best products in the world [9,23,24,41], which can be downloaded from https://ladsweb.modaps.eosdis.nasa.gov/ (accessed on 15 November 2022). Since its release, its algorithm has been updated many times, and the latest version is C6.1. In this study, MYD04 DT&DB combined AOD is used for comparison due to the near pass time (within 10 min) of OMI.

CALIPSO is a solar-orbiting Earth reconnaissance satellite of NASA and CNES. The active CALIOP sensor emits laser pulses at wavelengths of 532 nm and 1064 nm, and the acceptance channels measure the backscatter and orthogonal polarization signals at 532 nm and the backscatter signals at 1064 nm, respectively. By collecting information

on the attenuation scattering of aerosols and clouds on both bands and the polarized backscattering of 532 nm, the vertical profile of clouds and aerosols and the physical properties of cloud–aerosol particles are obtained. Different cloud and aerosol products are retrieved for different purposes; the Vertical Feature Mask (VFM) product of 532 nm is a level-2 product designed to classify atmospheric features into clouds or aerosols using the scene-classification algorithm [63].

AERONET is a ground-based aerosol remote-sensing observation network jointly established by NASA and loa-photons (CNRS) [64]. The network now covers major world regions, with more than 1500 sites worldwide, using the CIMEL Automatic Solar Photometer (SPAM) as the basic observation instrument. The AOD observation error of AERONET is about 0.01–0.02, and the AERONET data are often compared with multiple satellites to validate and refine their aerosol algorithms [42]. AERONET AOD data have 7 narrow bands with central wavelengths of 340, 380, 440, 500, 675, 870, and 1020 nm, with three data quality levels: 1.0 (unscreened), 1.5 (cloud-screened), and 2.0 (cloud-screened and quality assured). These data can be downloaded at https://aeronet.gsfc.nasa.gov/ (accessed on 15 November 2022). Sun–sky radiometer Observation NETwork (SONET) is a Chinese regional network similar to AERONET, and its data accuracy is equivalent to the level-1.5 data of AERONET (http://www.sonet.ac.cn/ accessed on 15 November 2022) [65,66].

### 2.2. Combination of Strategies and Evaluation Criterion

Unified spectral and spatiotemporal strategies are required for the study of Asia. Combining different datasets (ground-based, site-specific observations from AERONET; line traces for CALIPSO; and spatial data from OMI and MODIS) can help develop this strategy.

In this study, we collected all available data from 2006 to 2021, as the main satellites Aqua MODIS, OMI, and CALIPSO have provided products since 2002, 2004, and 2006, respectively, and AERONET has been operating since the last century. Even for polar-orbiting satellites that cover the globe once a day, the spatial distributions of monthly or seasonal means are often unsatisfactory due to the influence of clouds, resolution, and retrieval algorithms (especially OMI, due to abnormal pixels, and CALIPSO, due to line traces). The long-term average removes the accidental features and improves the overall features and spatial coverage. To obtain the before-and-after changes, this study divides the data into two time periods: 2006 to 2013 and 2013 to 2021. For statistical significance, only a pixel with data numbering >5 was used to compute the averages.

For unified spectrum, CALIPSO VFM product mainly classifies information in the vertical direction and does not involve band conversion. MODIS, OMI, AERONET, and SONET should be analyzed in the same band. From the description in 3.1, 500 nm is the band with the lowest amount of wavelength conversion; only the MODIS AOD needs to be derived from 440 and 550 nm using the following Ångström formula [67]. $\tau$ represents AOD at different bands, 440, 500, and 550 nm, and $\alpha$ represents the Ångström Exponent derived from known parameters.

$$\tau_{500}^{\text{MODIS}} = \tau_{440} \times \left(\frac{440}{500}\right)^{\alpha}; \; \alpha = \frac{ln\,\tau_{440}/\tau_{550}}{ln\,550/440} \tag{1}$$

In terms of unified spatiotemporal resolution, data from Aqua-MODIS are similar to OMI's overpass time; they can be easily resampled to a $1° \times 1°$ grid by using the best-quality flag [52,54]. The measurements from ground-based sites can be averaged within $\pm 1$ h of the satellite overpass. For CALIPSO, the VFM product provides a vertical profile with 0.03 km intervals from $-0.5$ km to 8.17 km altitude; the original swath data were gridded to a resolution of $1° \times 1° \times 0.03$ km to obtain the three-dimensional latitude–longitude–height matrix. According to the steps proposed in [39,40], the total samples and subtypes (dust and polluted dust, smoke, and elevated smoke) can be summarized to finally calculate the aerosol FoO in the selected Region Of Interest (ROI), while the data below 1.5 km altitude are not used due to large potential retrieval uncertainties.

This study was mainly concerned with the AAOD, which contained both aerosol magnitude and absorption information. However, previous studies illustrated that AOD simultaneously retrieved from OMI shows underestimation, which leads to inaccuracy in the AAOD. After spectral and spatiotemporal matching of MODIS and OMI data, importing MODIS AOD is a relatively simple way to improve the accuracy of AAOD. Formula (2) is used here, called MODIS-OMI AAOD in this study, and will be verified later. $\omega$ represents OMI SSA in each matched $1° \times 1°$ grid. Finally, all processed data are listed in Table 1 for easy viewing.

$$\text{AAOD}_{\text{MODIS−OMI}} = \omega_{500}^{\text{OMI}} \times \tau_{500}^{\text{MODIS}} \tag{2}$$

**Table 1.** The parameters used in this study.

| Parameters | Spectral Band | Resolution |
| --- | --- | --- |
| Aqua MODIS AOD | 500 nm | $1° \times 1°$ |
| OMI AAOD | 500 nm | $1° \times 1°$ |
| MODIS-OMI AAOD | 500 nm | $1° \times 1°$ |
| CALIPSO VFM | - | $1° \times 1° \times 0.03$ km |
| The smoke/dust frequencies from OMI | - | $1° \times 1°$ |
| AERONET/SONET AOD, AAOD | 500 nm | Within ±1 h of the satellite overpass |

In addition, when comparing satellite and terrestrial products, the Pearson correlation coefficient (R) is often used to describe the correlation between two datasets. If the R is above 0.8, it is generally considered a strong correlation, and below 0.4 is considered a weak correlation [22,23,41]. In this study, this coefficient was used to explore the correlation between observations at the center point AERONET and SONET sites, and AODs retrieved by satellites at different distances from the center point.

## 3. Results and Discussions

### 3.1. The Frequency Comparison of the Two Periods

AOD is a key parameter to characterize the total amount of aerosol, which is also the most commonly used parameter in time-series analysis [23,58]. The overall variation can be effectively obtained by comparing the means of 2006–2013 and 2014–2021. However, the change in aerosol types (mainly dust and smoke) during the two phases is more important.

Figure 1a,b shows the seasonal AOD mean of MODIS in two periods, 2006–2013 and 2014–2021. The average concentration of aerosols in eastern China decreased significantly in all seasons, especially in summer, about 0.4~0.7. These improvements account for the remarkable results of China's long-term air pollution management [14,68,69]. However, for Northeastern India, AOD increased in all seasons, and even the background value in the whole Indian region increased, which is related to India's rapid economic development [70]. Southeast Asia exhibits a strong seasonality change with stable high AOD during agricultural combustion in spring during two periods. Southeast Asia showed strong seasonal changes during agricultural burning in spring, and AOD was stable and high in two periods. Therefore, several ROIs with high AOD can be identified from the graph above: East China, Northeastern India, and South Asia, which is consistent with other studies [45,47]. In other places, due to the low average concentration, the change in aerosols is relatively insignificant.

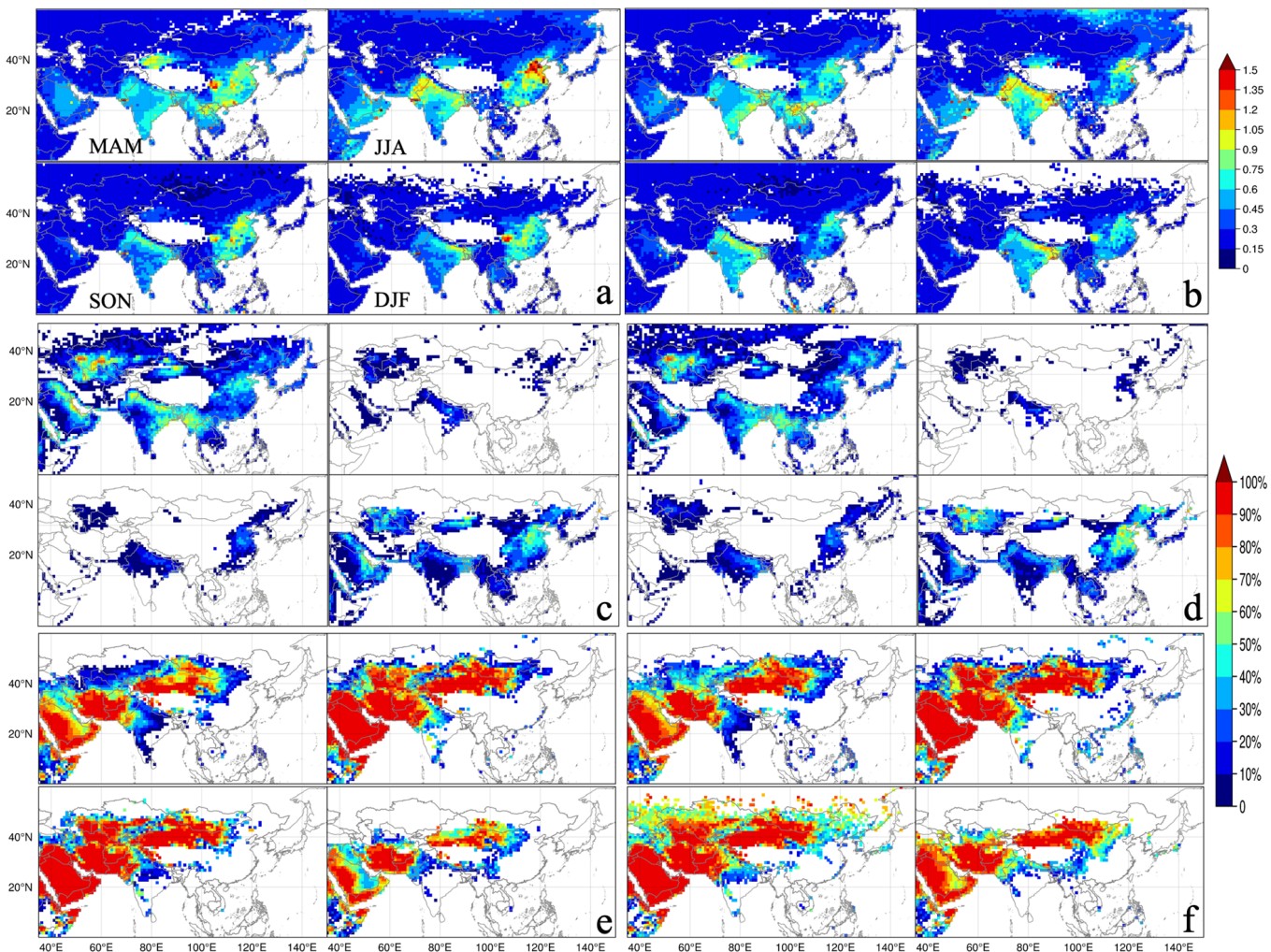

**Figure 1.** The season means of AOD at 500 nm from MODIS and the frequency of OMI smoke and dust type from OMI for two periods from 2006 to 2021: (**a**) MODIS means from 2006 to 2013; (**b**) MODIS means from 2014 to 2021; (**c**) The frequency of OMI smoke from 2006 to 2013; (**d**) The frequency of OMI smoke from 2014 to 2021; (**e**) The frequency of OMI dust from 2006 to 2013; (**f**) The frequency of OMI dust from 2014 to 2021.

Further, considering the aerosols of different types, Figure 1c,d shows the frequency of smoke type from OMI OMAERUV product, and Figure 1e,f represents the dust type. For each season, the occurrences of different types were counted and divided by the valid number of all aerosol occurrences in a single pixel. The classification of simple inversion parameters based on UVAI and VIS-AI has higher reliability than the retrievals from predefined assumptions [36]. The result is quite different from the trend of AOD. For smoke, human activity is a direct factor. Figure 1c,d shows that the frequency of smoke in all ROIs decreased in spring, which is associated with a greater reduction of straw burning in agricultural activities, although the occurrence probability of smoke in Southeast Asia and the Indian plains is still 40~50%. In winter, although the total amount of AOD after 2013 was lower than that in the period before 2013, the occurrence probability of smoke rose to 70~80% for Eastern China, which thus became the main area of absorbing aerosols. The change in aerosol type is related to the adjustment of heating mode and industrial structure in winter. Figure 1e,f shows the distribution of dust from a natural source, and an interesting phenomenon is that the dust spread from west to east, and the high-frequency area and the affected area expanded in all seasons. In spring and winter, due to the dry and monsoon climate, sand and dust spread easier, and the strong sand and dust aerosols in Xinjiang and Inner Mongolia accounted for more than 70% and spread to northern

China. The distribution in the latter period shows that the frequency in the southern region also greatly increased. In summer, the humidity in the atmosphere was high, and the monsoon effect was insignificant, so the high-frequency area was relatively stable. In autumn, although the high-frequency region was relatively stable, the dust frequency increased by 10% throughout western and northern Asia. The highest value appeared in the Beijing–Tianjin–Hebei region in May because the region was affected by agricultural production in spring (agricultural production time lags behind Southeast Asia due to different latitudes), dusty weather in the north, and local industrial production emissions. In addition, the data before 2013 account for a large proportion.

In summary, over a longer period, AOD in eastern China decreased in all seasons, but the frequency of absorbing aerosol in winter increased. The Indian subcontinent represents a hotspot for mixed scattering aerosol, with the proportion of pure absorbing aerosol prominent during spring. In Western Asia, Mongolia, and other places dominated by dust aerosols, the AOD remained low. In terms of the frequency of occurrence, there was a trend of eastward movement, and the area expanded in all seasons.

### 3.2. Comparison of Smoke AAOD between Two Periods

AAOD represents the concentration of absorbing aerosols; however, its accuracy decreases dramatically with the decrease in aerosol concentration. AERONET recommends that AAOD/SSA has high credibility when the AOD at 440 nm is greater than 0.3 or 0.4. Meanwhile, with a smaller AOD, the uncertainty increases to two to three (https://aeronet.gsfc.nasa.gov/new_web/PDF/AERONETcriteria_final1.pdf, last access 14 July 2022). Here, only the AAOD of smoke-type data with AOD > 0.3 was selected (due to the low dust AAOD, there is no significant change in the mean value, so it is not listed). Figure 2a,b shows the OMI AAOD, and Figure 2c,d shows the MODIS-OMI AAOD at 500 nm.

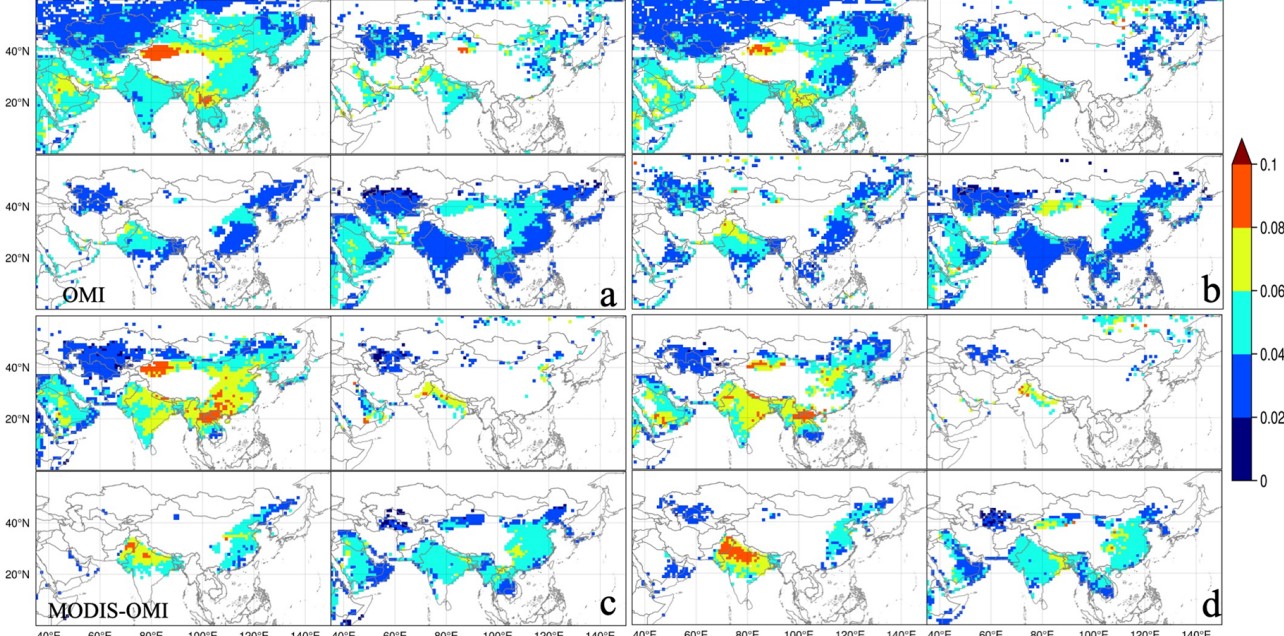

**Figure 2.** The seasonal smoke AAOD means at 500 nm from OMI and MODIS-OMI for two periods from 2006 to 2022: (**a**) OMI AAOD from 2006 to 2013; (**b**) OMI AAOD from 2014 to 2021; (**c**) MODIS-OMI AAOD from 2006 to 2013; (**d**) MODIS-OMI AAOD from 2014 to 2021.

A significant decline in area and concentration (more than 0.2) can be seen in Southeast Asia in spring, indicating a reduction in burning events caused by agricultural production. The northern Indian plain in autumn became a new high-value area for absorbing aerosol. In the autumn of 2013, the area with high AAOD expanded by about 100% compared with the previous period before 2013, and the concentration increased by 30%, which is directly related to the industrial development of India. From the perspective of seasonal trends, the absorbing aerosols in summer are the lowest in China due to the high temperature and humidity in summer, and the aerosols are prone to hygroscopic growth and thus show strong scattering. Next is autumn, followed by winter and spring, with winter most impacted by the accumulation of absorbing aerosols spurred on by heating and calm weather. However, spring is also affected by agricultural activity. In spring, the concentration and area of AAOD in eastern China after 2013 decreased compared to before 2013, which is consistent with the change in frequency in Figure 1. The AAOD value increased slightly in winter, but it was not significant, which may be because of the homogeneity of the average data.

It can be seen that the overall trend of OMI and MODIS-OMI AAOD data is consistent, which verifies that they are roughly correct. The addition of MODIS AOD improves the spatial details and captures finer signatures in some areas. Considering the accuracy of MODIS AOD in global verification, it is reasonable to think that the AAOD from combined MODIS-OMI with the same spatiotemporal scales can better represent the real situation, which has been validated in some studies [52,54,58]. In addition, high amounts of smoke aerosols appear in Xinjiang in spring, which may be due to the classification error of dust and smoke with small AOD (refer to OMI-OMAERUV algorithm ATBD document), which also needs further demonstration.

### 3.3. Time-Series Comparison

The monthly means showed more continuous and detailed change characteristics for the aerosols in Asia; Figure 3a shows the variation from MODIS-OMI AAOD. Southeast Asia has a concentrated high value in March and April, while there are not even enough pixels of absorbing aerosol for calculation in other months, indicating that the region is mainly dominated by biomass combustion aerosol produced by agriculture, which also appeared in March and April in southern India. However, high amounts of absorbing aerosols appeared in all months except during summer in northern India near the Himalayas because most industrial cities in northern India are concentrated. The Himalayan barrier makes it difficult for the aerosol to disperse. In April, the aerosol concentration was affected by agricultural and industrial production and reached the highest value. At the same time, eastern China is also the peak area for absorbing aerosols, especially in the northern regions with developed industries, such as Beijing–Tianjin–Hebei and the Shandong province. The highest value appeared in the Beijing–Tianjin–Hebei region in May because the region was affected by agricultural production in spring (the agricultural production time lags behind Southeast Asia due to different latitudes), dust weather in the north, and local industrial production emissions. In addition, the lax straw-combustion control policy before 2013 makes this part of the data more prominent, resulting in the overall high value. In addition, the dust aerosol in Xinjiang in spring also led to significantly high AAOD values in March and April.

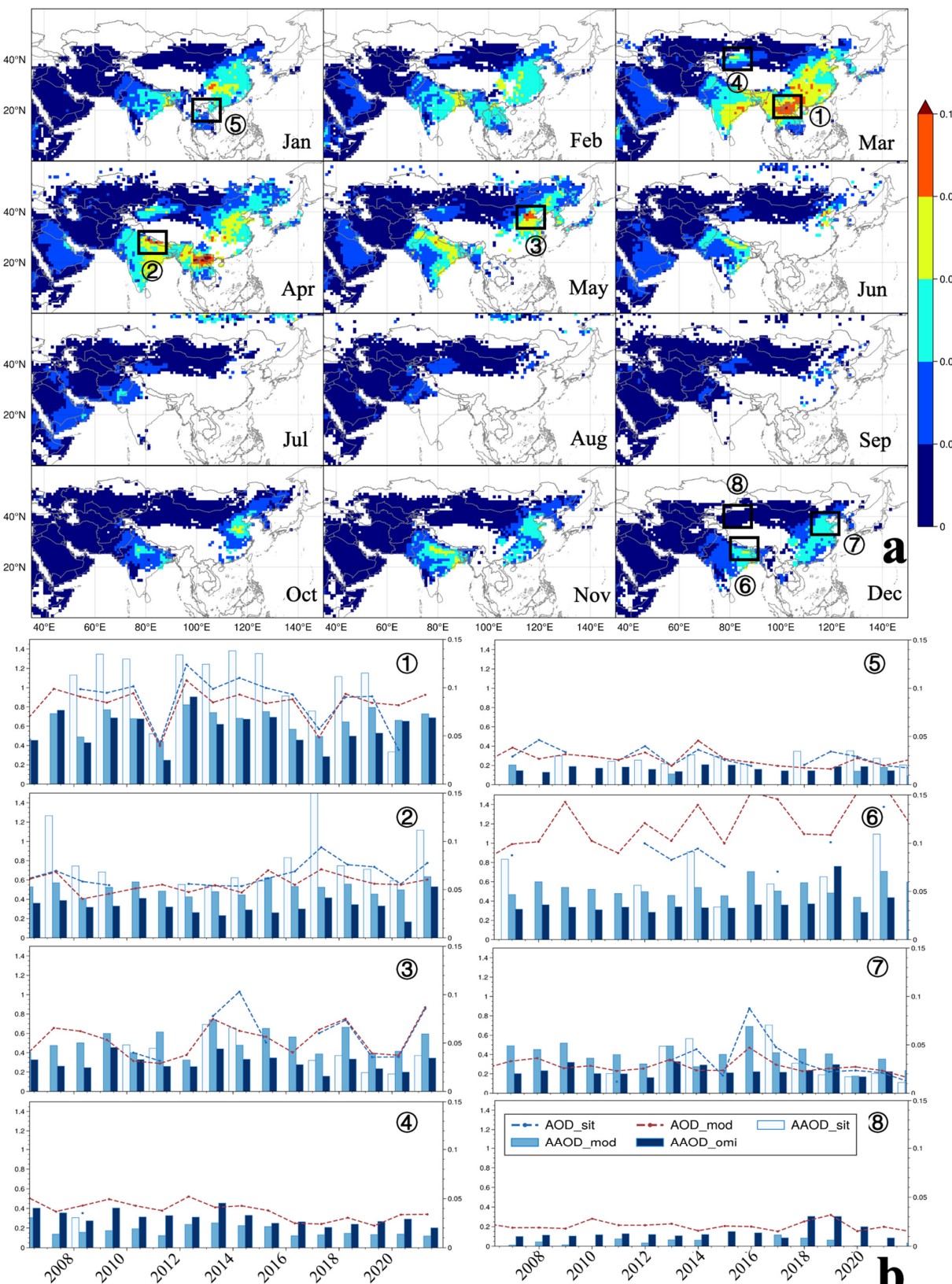

**Figure 3.** (**a**) Monthly AAOD from MODIS-OMI from 2006 to 2021; (**b**) the time-series variations of AERONET AOD (dotted blue line), MODIS AOD (dotted red line), AERONET AAOD (white column), MODIS-OMI AAOD (light blue column), and OMI AAOD (blue column) at the corresponding month.

Then, we chose four ROIs to compare the annual change trend of AAOD under two different months (Figure 3b). Among them, no. 1~4 are AERONET sites with high values of continuity in the four selected regions: the Chiang_Mai_Met_Sta site in Southeast Asia, the Gandhi_College site in northern India, the Beijing_RADI site in Beijing–Tianjin–Hebei, and the Zhangye site in Xinjiang; no. 5~8 correspond to the previous region but represent the low-value month. Southeast Asia shows the most dramatic differences in different months. In March and April, the average value of AOD exceeds 1, and the ground-based AAOD is higher than 0.1, while in other months, the average value of AOD is less than 0.4, and the AAOD is less than 0.03. Northern India has a small mean AOD value (0.6 to 0.8) and a large AAOD (higher than 0.06) in spring, while the AAOD decreases slightly in winter, but the AOD increases to more than 1, which indicates that the background pollution in this region is sustained at a high value. In spring, it is polluted by agriculture, while in winter, the aerosol pollution is more serious, mainly due to scattering pollution. The pollution in Beijing is generally at a low value. In recent years, AOD and AAOD in winter have shown a downward trend, while in spring, the pollution is affected by sand and dust, and the interannual change is large. Xinjiang is mainly dominated by dust aerosols, with the highest AAOD value of about 0.04 in spring (there are few foundation data at the station). A significant feature is that the AAOD value of OMI was higher than that of MODIS-OMI, indicating that the AOD in the OMI algorithm is higher than that of MODIS. This may be caused by different assumptions about the height of the sand dust layer in the MODIS and OMI algorithms, so more research is needed in the future. In general, in areas with high AAOD, MODIS-OMI AAOD is closer to the AAOD measured on the ground. However, due to the complexity of aerosol types, the AAOD obtained by different methods varies greatly.

In addition, we can see obviously low values in 2011 (especially in Chiang Mai) in Figure 3b. Some papers have speculated that this may be related to the changes in El Niño/Southern Oscillation (ENSO), North Atlantic Oscillation (NAO), and Indian Ocean Dipole (IOD) (these three indexes have a high correlation with the human combustion of biomass emissions, and they all have low values in 2011) [71,72].

*3.4. Regional Relevance*

The time-series comparison of a single site provided the annual change trend in a fixed area, but the spatial change information is more important for aerosol traceability and transport analysis. Aerosols move and propagate in the atmosphere under the combined influence of wind and terrain, and some aerosols travel long distances. In the Northern Hemisphere aerosol is dispersed eastwards because of the Hadley circulation patterns [73]. Using the daily mean value of AOD observed at the station and the surrounding pixels for correlation analysis, we can preliminarily judge the diffusion direction and range of aerosols.

Figure 4 shows the distribution of the correlation coefficient in three different cases. Panels a and b show a relatively flat terrain and stable atmospheric environment. They are a roughly circular area with 6° latitudes and longitudes, with a significant correlation (R > 0.4). In particular, Mongolia is affected by monsoons all year, covering Beijing and other places downstream (which proves the situation of the Beijing station in Figure 3), but has no impact upstream. The situation of panels c and d is significantly affected by the terrain. The impact range is a strip from west to east, especially in northern India, which is blocked by the Himalayas, and the correlation between east and west is significant. Panels e and f show a big surrounding region with a strong correlation to the AERONET sites. This may be because the local agricultural activities and industrial structure are similar, and the occurrence of high-concentration aerosol is close, resulting in the same dominant aerosol types. This contributes to the improvement in the aerosol-type retrieval algorithm.

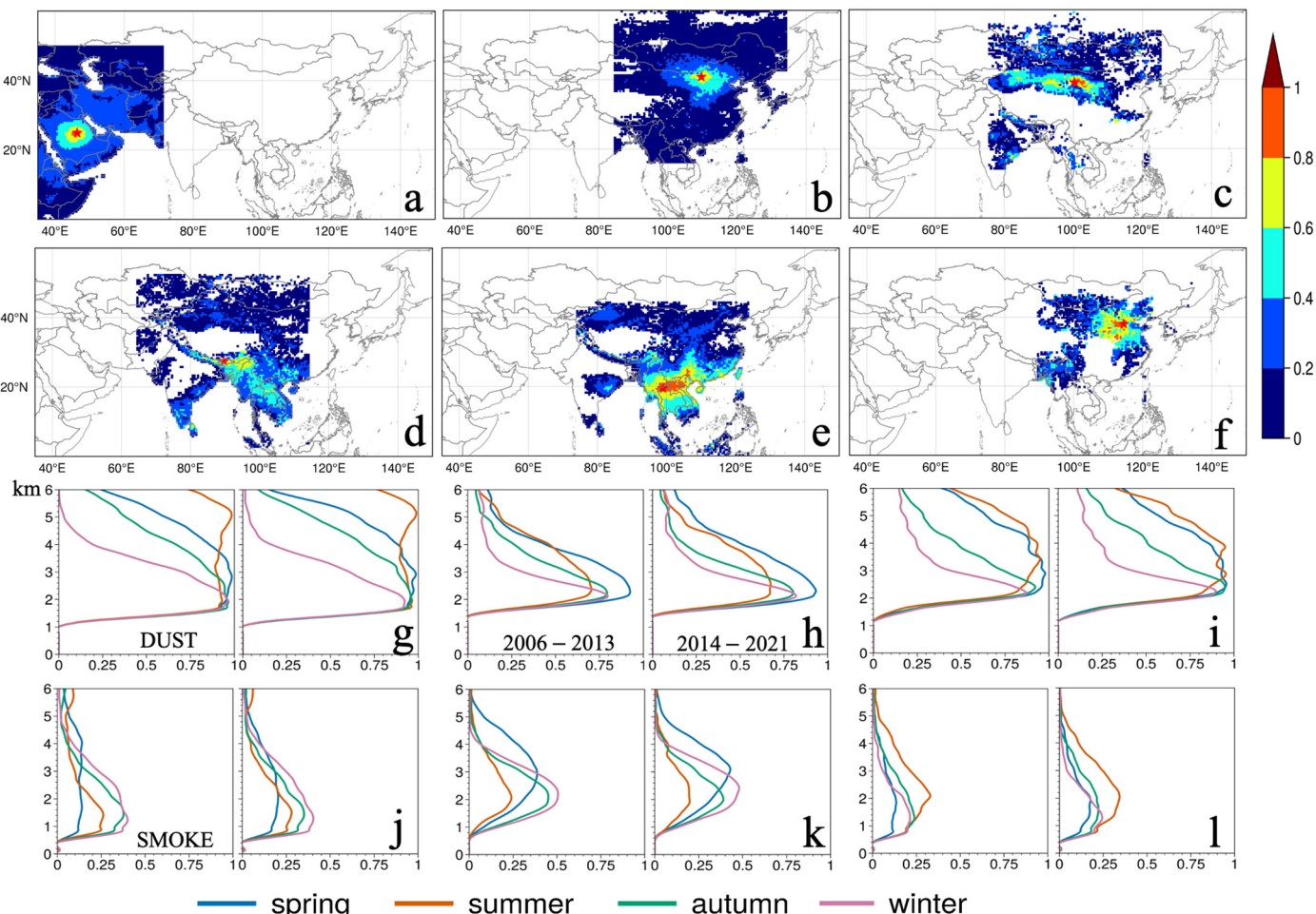

**Figure 4.** The correlation coefficient (R) between MODIS and AERONET daily AOD of each pixel near the sites: (**a**) Solar_Village site; (**b**) AOE_Baotou site; (**c**) Zhangye site; (**d**) Thimphu; (**e**) Son_La site; (**f**) Shijiazhuang-SZF; (**g**–**l**): seasonal vertical profiles of FoO with two aerosol types (dust and smoke) and different season (lines of different colors) during two periods over the area with R > 0.6 for the same sites as panels (**a**–**f**); *x*-axis represents the normalized frequency from 0 to 1, and *y*-axis represents the altitude from 0 to 6 km.

In the Arab region, dust aerosols are dominant. Except in summer, the highest values in other seasons are distributed at an altitude of 2~3 km. In summer, due to the turbulence caused by higher temperatures, dust aerosols are evenly distributed at 2~5 km. Inner Mongolia is affected by the northern monsoon. The proportion of sand dust in spring is higher than 75%, and the dust-raising height in the four seasons is relatively stable. Dust aerosol lifts higher over the Gobi of Xinjiang in spring and summer (3–4 km) than in autumn and winter (2–2.5 km).

In terms of the comparison of smoke aerosols in India, Southeast Asia, and eastern China, the aerosol-accumulation layer in India is low, reaching the maximum value at 1~1.5 km (uniformly distributed within 1~5 km in spring due to agricultural burning), which is consistent with previous studies [74]. In Southeast Asia, except in spring, the highest value in other seasons is about 2 km. In spring, there is not only a large amount of aerosols but also a high altitude of 3~5 km. Eastern China is different; thanks to strict policy control, the proportion of smoke in spring is not prominent. On the contrary, in summer, photochemistry intensifies due to a rise in temperature, with smoke accounting for a relatively high proportion and rising to 2 km.

To further explore the changes in the vertical distribution of dust and smoke aerosols, Figure 5 shows the altitude-resolved FoO profiles with 5–10° longitudinal intervals during

2006–2013 and 2014–2021. Several natural hotspots, such as deserts and arid areas with strong aerosol emissions, can be easily highlighted in Figure 5a,b. The Tibetan Plateau has a good barrier effect on dust from west to east, with a high possibility of dust occurrence in the south and mixing with local background aerosol, resulting in complex aerosols in India and other places. The dust aerosols in northern and eastern China are mainly from Xinjiang and Mongolia due to a height drop from north to South at lon. 110° and 120°. It can be seen that the aerosol changes in the two time periods are insignificant, except for an increase in Japan in the latter stage (lon. 140°). It is worth mentioning that polluted dust, as a type of CALIPSO classification, may include a mixture of dust and urban pollutants or just dust-like particles generated by the transformation of hygroscopic growth [33].

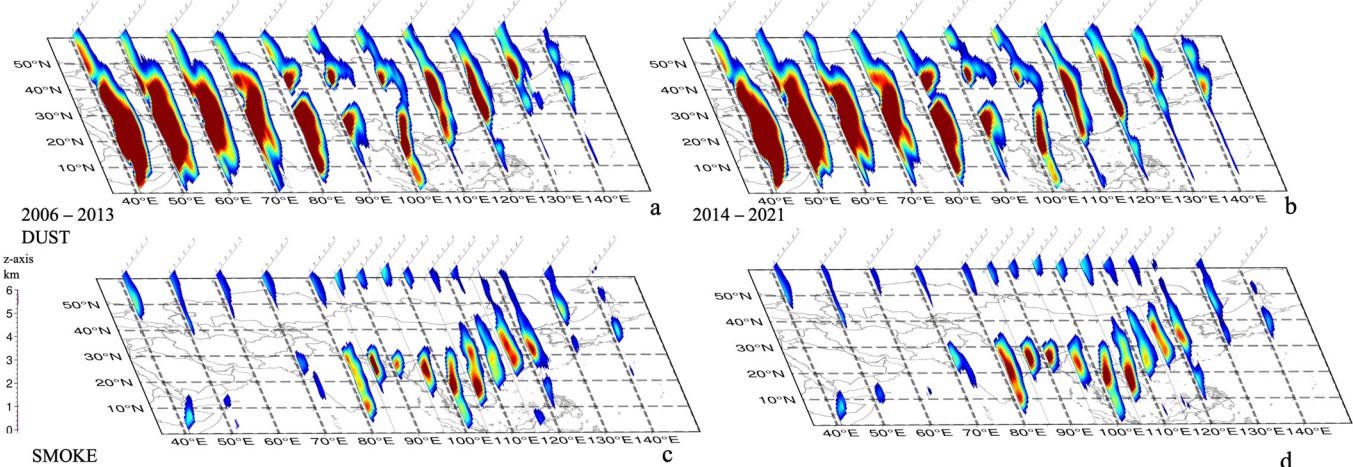

**Figure 5.** Averaged latitude–vertical 3D aerosol FoO (>5%) derived from level-2 CALIOP VFM products during two periods, 2006–2013 and 2014–2021, along longitudes from 40°E to 140°E (the interval is 5 or 10 degrees). (**a**) DUST FoO from 2006 to 2013; (**b**) DUST FoO from 2014 to 2021; (**c**) SMOKE FoO from 2006 to 2013; (**d**) SMOKE from 2014 to 2021.

Southeast Asia, India, and eastern China are known for anthropogenic aerosol emissions due to agricultural and industrialized activities Figure 5c,d. Southeast Asia (lon. 105°) still has the consistently highest value, but more smoke is concentrated locally, and the phenomenon of northward propagation is weakened in the later stage of 2014–2021. In eastern China, the distribution of vertical smoke FoO shows that the frequency and height of smoke (lon. 110~120°) have been weakened, which is similar to the results from OMI (except in winter). Combined with the decline in total AOD from MODIS, this indicates that atmospheric pollution control in China has achieved some results. In India (lon. 80~100°), there is a significant increase in both frequency and height, especially around lon. 80°, which is the biggest change in this figure. This is related to the industrial development in India and needs further attention.

## 4. Conclusions

In this study, the data from OMI, MODIS, CALIPSO, and AERONET, with two sufficiently long periods of more than 8 years, were compared to clarify the spatial–temporal correlation, variability, and vertical structure change in absorbing aerosols, including dust and smoke. Overall, the total concentration of aerosol decreased significantly in eastern China and increased significantly in northern India in all seasons. However, the frequency of absorbing aerosol relating to human activities in eastern China increased significantly in winter, while in India, mixed scattering aerosols were dominant, and the absorptivity was not significant. In terms of the concentration of absorbing aerosol, the highest value occurs in Southeast Asia in spring and has decreased by more than 0.2. On the contrary, the northern Indian plain in autumn became a new growth point of absorptive aerosols, both in concentration and frequency. In most areas of anthropogenic aerosol, MODIS

AOD is in good agreement with ground-based observation results, while OMI AAOD is significantly underestimated. After the introduction of MODIS AOD, the numerical value of MODIS-OMI AAOD increased, and the spatiotemporal distribution performance improved. However, in dust-dominated areas, OMI AAOD is higher than MODIS-OMI ones. Due to the lack of foundation data verification, no further suggestion can be made. The spatial correlation of aerosols is related to aerosol concentration, topography, and climate. In the spring of Southeast Asia, there is a higher peak height of smoke at 3 km and a greater vertical range from 1 to 5 km (the peak heights of eastern China and India are 1~2 km), which affects Taiwan as far as possible, covering a 20° lon.–lat. grid. This is related to strong trade winds of the tropics, which inject some smoke aerosol into the free troposphere and spread eastward to large areas [75]. The distribution of vertical smoke FoO from CALIPSO shows that the frequency and height of smoke in eastern China (lon. 110~120°E) have been weakened, similar to the results from OMI (except in winter). On the contrary, there is a significant increase in India (lon. 80~100°E), both in frequency and height, especially around lon. 80°E. However, this is different from the OMI classification result, possibly due to the higher accuracy of active remote sensing.

In addition, for dust from natural sources, we should also be alert to eastward expansion (OMI result), although there was no significant difference from the FoO height of CALIPSO from 2006 to 2021. Dust in different regions also has certain differences. It is usually distributed at an altitude of 2–5 km. The heat-convection effect in summer leads to a higher lifting height, while the dust in Inner Mongolia is more significant in spring, and the peak value is about 2–3 km (lower than in other regions), which may be related to the perennial monsoon effect.

**Author Contributions:** D.L. designed the study and wrote the manuscript, and Y.X. and K.Q. contributed to the preparation of the manuscript through review, editing, and comments. H.W. and L.W. proposed many useful suggestions to improve its quality. H.K. contributed to the writing of the content. All authors were involved in modifying the paper, the literature review, and the discussion of the results. All authors have read and agreed to the published version of the manuscript.

**Funding:** This research was funded by the Fundamental Research Funds for the Central Universities, under grants 2020CXNL08 and 2021QN1033, and the National Natural Science Foundation of China (NSFC) under grants 42075132 and 41871260.

**Data Availability Statement:** Not applicable.

**Acknowledgments:** The MODIS, OMI, and CALIPSO data used in this study were also acquired from the Goddard Earth Sciences (GES) Data and Information Services Center (DISC) and Distributed Active Archive Center (DAAC). Thanks for the public data provided by the official team. We would also like to thank the AERONET and SONET projects and the data from different sites operated and maintained by local researchers in this work. We gratefully acknowledge the anonymous reviewers whose feedback contributed to improving the manuscript.

**Conflicts of Interest:** The authors declare no conflict of interest.

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
