# Peer review of "Investigating the Long-Term Variation Trends of Absorbing Aerosols over Asia by Using Multiple Satellites"

_remotesensing, doi:10.3390/rs14225832_

Round 1
Reviewer 1 Report
This paper reports analysis of absorbing aerosols over Asia using data from satellite sensors OMI, MODIS, CALIOP and ground-based AERONET. Distribution and long-term variation of light absorbing aerosols including mineral dust are analyzed, and interesting results are presented.
It seems the standard data sets from OMI, MODIS, and CALIOP are mostly used, but it is not clear about the MODIS-OMI AAOD, which is shown in Figure 2 (c, d). If it is a new data set, details of the derivation method should be described. The descriptions in the text (lines 242-245) are not sufficient.
The manuscript is generally well written except for the above mentioned point. However, some of descriptions are not very accurate. Revisions are recommended.
Specific comments
line 16: What is “what?”?
line 21 “low dust aerosol”: It is not clear what is low. Distribution height?
line 21 “transmission path” : Is term “transmission” appropriate for aerosol transport?
line 93: “decomposition”? —> composition(?)
line 106: “Therefore”? —> However(?)
lines 137-140: The description is not accurate. CALIOP doesn’t have polarization channels at 1064 nm. Polarization components measurement is only at 532 nm. Also, the term “quadrature polarization” is not common. It would be better to use “orthogonal polarization”.
line 170: OMI is since 2004, and CALIPSO is since 2006.
lines 244-255: Detailed description needed.
line 392: “T” ?
line 329: Definition of correlation coefficient should be explicitly described.
line 358: “relative coefficient”?
line 373: “(mixed with water vapor and converted into similar properties) “: It is an inaccurate description. The definition of “polluted dust” in the CALIPSO product should be explained.
Refelence 20 and 47 are the same.
Author Response
Thank you for your comments to help me finish my manuscript better. We have completed the following items: (1) Revised the English grammar of the whole manuscript; (2) Rewritten many chapters including abstract, method and summary to build better logic; (3) Enlarged the axis labels and increased the unit in figures. The following is my point-to-point reply. Please read the content in the revised version (used red color to mark) for more details.

Reviewer 2 Report
1: This article needs extensive editing to fix the grammar and tenses which must be done before a proper scientific review can be conducted.
2: I question if there is sufficient originality in this work. MODIS, CALIPSO, and AERONET combinations are regularly investigated.
3: The authors make a number of unsubstantial statements take line 28 for example absorbing aerosols are undefined. 10% is quoted from where? What is the balance of the aerosol? The entire document needs to be rewritten asking is something defined, is something quoted, what does the first clause mean to "the problem"?
4: The abstract needs to be totally rewritten, statement of the problem, work and conclusions.
Author Response

(The authors gave the same response as above.)

Reviewer 3 Report
Comments to the author:
This manuscript (remotesensing-1955590) compared AOD results for Asia from several datasets, including OMI, MODIS, CALIPSO and AERONET. This study provides an integrated summary of the AOD and AAOD evolution during the last decade in Asia.
Major comments:
1) Line 16. “the peaks of what? are concentrated around 1.5-2 km in other regions”. Please check the manuscript carefully before submission.
2) Line 170-171. Please check the font settings.
3) As for Figure 1, the authors stated that “AOD in eastern China decreases”. It would be difficult for the readers to derive this statement from Figure 1. It would be helpful to include time-series plots of AOD of specific areas (similar to Figure 3).
4) Line 230. “The season mean” should be “The seasonal means”
5) Line 271. “The season smoke AAOD means” should be “The seasonal smoke AAOD means”
6) As shown in Figure 3b, the time-series plot of region â‘ demonstrated a significant drop in 2011, any explanations?
7) Figure 4. Please specify the y-axis and x-axis titles, as well as units for the vertical profile plots.
8) It is suggested to compare the AAOD trends with ground-based long-term black carbon measurements (Zhang et al., 2019; Kanaya et al., 2020; Sun et al., 2022), to see how well can the agreement could be for the long-term trends.
References
9) Kanaya, Y., Yamaji, K., Miyakawa, T., Taketani, F., Zhu, C., Choi, Y., Komazaki, Y., Ikeda, K., Kondo, Y., and Klimont, Z.: Rapid reduction in black carbon emissions from China: evidence from 2009–2019 observations on Fukue Island, Japan, Atmos. Chem. Phys., 20, 6339-6356, doi: https://doi.org/10.5194/acp-20-6339-2020, 2020.
10) Sun, J., Wang, Z., Zhou, W., Xie, C., Wu, C., Chen, C., Han, T., Wang, Q., Li, Z., Li, J., Fu, P., Wang, Z., and Sun, Y.: Long-term changes in black carbon and aerosol optical properties from 2012 to 2020 in Beijing, China, Atmos. Chem. Phys., 22, 561-575, doi: https://doi.org/10.5194/acp-22-561-2022, 2022.
11) Zhang, Y., Li, Y., Guo, J., Wang, Y., Chen, D., and Chen, H.: The climatology and trend of black carbon in China from 12-year ground observations, Climate Dynamics, 53, 5881-5892, doi: https://doi.org/10.1007/s00382-019-04903-0, 2019.
Author Response

(The authors gave the same response as above.)

Round 2
Reviewer 1 Report
line 362-364: El Niño/Southern Oscillation (ENSO), North Atlantic Oscillation (NAO), Indian Ocean Dipole (IOD) should be spelled out. Also, the meaning of the sentence in the bracket is not clear. It means 2011 was a La Niña year, right?
Author Response
We have made the following modifications in this version: (1) use editing services to improve English grammar; (2) Rewrite the abstract according to the comments from reviewer2 (I appreciate your help so much; we have gained valuable experience on how to write a good abstract); (2) Some explanations have been added to the conclusion.
Thanks to all the experts again.
_______________________________________
The response:
Yes, it is a La Niña year, we consulted the professor who did relevant researches and looked up some literatures.
Reviewer 2 Report
The grammar remains poor. I have rewritten the abstract below. Please fix the English before sending out articles for scientific review.
Absorbing aerosols, consisting of smoke (Black Carbon (BC) and other organics) and dust (from windblown sources) can have a strong warming effect on the climate and impact atmospheric circulation due to localized heating. To investigate the spatiotemporal and vertical changes of absorbing aerosols across Asia the collocation data from OMI, MODIS and CALIPSO were used to compare two time periods of 2006-2013 and 2014-2021. This study revealed a significant temporal and spatial contrast of aerosol loading over the study region with a drop in aerosol concentration recorded across the Eastern China region and a concurrent increase over the Indian sub-continent region. The range of aerosol diffusion is affected by the height of the smoke and aerosol plumes and the wind force. Smoke from Southeast Asia typically rises to a height of 3 km in contrast to other regions where it is found to be around 1.5-2 km) (reference this [and I don’t believe this is true]) and (explain this) (my guess) as the higher latitudes of smoke in Australia/Africa, and North America are closer to the strong trade winds of the tropics. In the Northern Hemisphere aerosol is dispersed eastwards because of the Hadley circulation patterns (ref this). The study showed that dust in Inner Mongolia had the lowest plume height of 2 km (typically in spring) compared to other locations across the study region) where it reached 2-5 km in summer. The study showed by comparison with AERONET measurements that combining data from MODIS and OMI generates more accuracy in detecting aerosol AOD from smoke than using the instruments singularly. This study has provided a comprehensive assessment of absorbing aerosol [something still missing here] in Asia by utilizing multiplatform remote sensed data and has summarized the long-term changes in the spatiotemporal distribution of aerosols, transmission [what do you mean by transmission] pathways, and vertical structure of absorbing aerosols.
Author Response
We have made the following modifications in this version: (1) use editing services to improve English grammar; (2) Rewrite the abstract according to the comments from reviewer2 (I appreciate your help so much; we have gained valuable experience on how to write a good abstract); (2) Some explanations have been added to the conclusion.
Thanks to all the experts again.

Reviewer 3 Report
The English of the manuscript still needs to be polished.
Author Response

(The authors gave the same response as above.)
